# Pruning Transformers with a Finite Admixture of Keys

## Abstract

Pairwise dot product-based self-attention is key to the success of transformers which achieve state-of-the-art performance across a variety of applications in language and vision, but are costly to compute. However, it has been shown that most attention scores and keys in transformers are redundant and can be removed without loss of accuracy. In this paper, we develop a novel probabilistic framework for pruning attention scores and keys in transformers. We first formulate an admixture model of attention keys whose input data to be clustered are attention queries. We show that attention scores in self-attention correspond to the posterior distribution of this model when attention keys admit a uniform prior distribution. We then relax this uniform prior constraint and let the model learn these priors from data, resulting in a new Finite Admixture of Keys (FiAK). The learned priors in FiAK are used for pruning away redundant attention scores and keys in the baseline transformers, improving the diversity of attention patterns that the models capture. We corroborate the efficiency of transformers pruned with FiAK on practical tasks including ImageNet object classification, COCO object detection, and WikiText-103 language modeling. Our experiments demonstrate that transformers pruned with FiAK yield similar or better accuracy than the baseline dense transformers while being much more efficient in terms of memory and computational cost.

## 1 Introduction

Transformers Vaswani et al. (2017) have been becoming the method of choice in computer vision and machine learning Al-Rfou et al. (2019); Dai et al. (2019); Williams et al. (2018); Devlin et al. (2018); Brown & et al. (2020); Howard & Ruder (2018); Rajpurkar et al. (2016); Dehghani et al. (2018); So et al. (2019); Dosovitskiy et al. (2020); Touvron et al. (2020b). Thanks to their ability to learn from unlabeled data and from different data modalities, transformers have achieved state-of-the-art performance on a wide range of tasks and applications, including image recognition, object detection, and language modeling Radford et al. (2018; 2019); Devlin et al. (2018); Yang et al. (2019); Liu et al. (2019). Lying at the heart of transformers is the self-attention mechanism, which captures the contextual representation of the input sequence by allowing each token in the input sequence to pay attention to other tokens Cho et al. (2014); Parikh et al. (2016); Lin et al. (2017); Bahdanau et al. (2014); Vaswani et al. (2017); Kim et al. (2017). In particular, self-attention represents each token as the weighted average of the other tokens' feature representation using the similarity scores between the tokens as weight coefficients. The capability of self-attention to attain diverse syntactic and semantic representations accounts for the success of transformers in practice Tenney et al. (2019); Vig & Belinkov (2019); Clark et al. (2019); Voita et al. (2019a); Hewitt & Liang (2019).

### 1.1 Self-Attention

Given an input sequence $\boldsymbol{X} = [\boldsymbol{x}_1, \ldots, \boldsymbol{x}_N]^\top \in \mathbb{R}^{N \times D_x}$ of $N$ feature vectors, the self-attention transforms it into another sequence $\hat{\boldsymbol{V}} = [\hat{\boldsymbol{v}}_1, \ldots, \hat{\boldsymbol{v}}_N]^\top \in \mathbb{R}^{N \times D_v}$ as follows

$$\hat{\boldsymbol{v}}_i = \sum_{j=1}^N \text{softmax}\left(\frac{\boldsymbol{q}_i^\top \boldsymbol{k}_j}{\sqrt{D}}\right) \boldsymbol{v}_j, \text{ for } i = 1, \ldots, N, \tag{1}$$

where the scalar $\text{softmax}((\boldsymbol{q}_i^\top \boldsymbol{k}_j)/\sqrt{D})$ can be understood as the attention $\hat{\boldsymbol{v}}_i$ pays to the input feature $\boldsymbol{x}_j$. The vectors $\boldsymbol{q}_i, \boldsymbol{k}_j$, and $\boldsymbol{v}_j$ are called the query, key, and value vectors, respectively; these vectors are computed

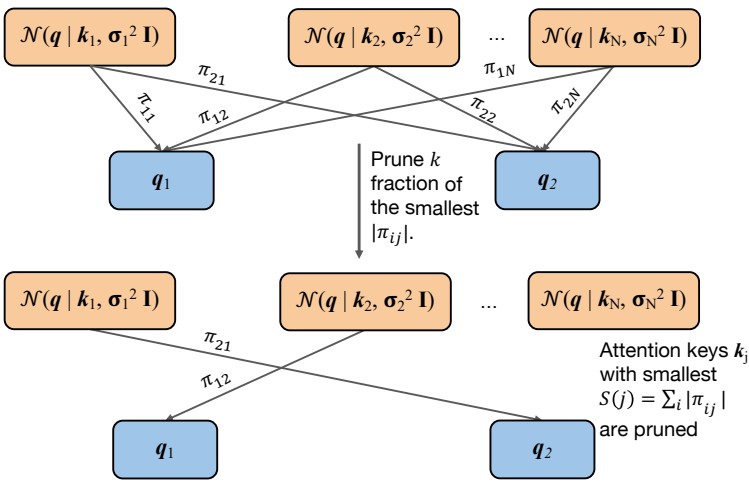

Figure 1: Our Finite Admixture of Keys (FiAK) models the distribution of the queries $\boldsymbol{q}_i$ in self-attention by an admixture model whose cluster components center around the attention keys $\boldsymbol{k}_j$, i.e. $p(\boldsymbol{q}_i) = \sum_{j=1}^{N} \pi_{ij}\mathcal{N}(\boldsymbol{q}_i \mid \boldsymbol{k}_j, \sigma_j^2 \mathbf{I})$, $i, j = 1, \dots, N$. The prior distributions $\pi_{ij}$ in the admixture are used to prune redundant attention scores $a_{ij} = \text{softmax}\left(\frac{\boldsymbol{q}_i^\top \boldsymbol{k}_j}{\sqrt{D}}\right)$. The scores $S(j) = \sum_i |\pi_{ij}|$ are used to prune redundant keys $\boldsymbol{k}_j$. A fraction of attention scores $a_{ij}$ and keys $\boldsymbol{k}_j$ with the smallest $|\pi_{ij}|$ and $S(j)$, respectively, will be pruned away to save memory and computation.

as follows

$$
\begin{aligned}
[\boldsymbol{q}_1, \boldsymbol{q}_2, \dots, \boldsymbol{q}_N]^\top := \boldsymbol{Q} = \boldsymbol{X}\boldsymbol{W}_Q^\top \in \mathbb{R}^{N \times D}, \\
[\boldsymbol{k}_1, \boldsymbol{k}_2, \dots, \boldsymbol{k}_N]^\top := \boldsymbol{K} = \boldsymbol{X}\boldsymbol{W}_K^\top \in \mathbb{R}^{N \times D}, \\
[\boldsymbol{v}_1, \boldsymbol{v}_2, \dots, \boldsymbol{v}_N]^\top := \boldsymbol{V} = \boldsymbol{X}\boldsymbol{W}_V^\top \in \mathbb{R}^{N \times D_v},
\end{aligned}
\tag{2}
$$

where $\boldsymbol{W}_Q, \boldsymbol{W}_K \in \mathbb{R}^{D \times D_x}$, and $\boldsymbol{W}_V \in \mathbb{R}^{D_v \times D_x}$ are the weight matrices. We can further write Eqn. 1 into the following compact form

$$
\hat{\boldsymbol{V}} = \text{softmax}\left(\frac{\boldsymbol{Q}\boldsymbol{K}^\top}{\sqrt{D}}\right)\boldsymbol{V} = \mathbf{A}\boldsymbol{V},
\tag{3}
$$

where the softmax function is applied to each row of the matrix $(\boldsymbol{Q}\boldsymbol{K}^\top)/\sqrt{D}$.

For each query vector $\boldsymbol{q}_i$ for $i = 1, \cdots, N$, an equivalent form of Eqn. 3 to compute the output vector $\hat{\boldsymbol{v}}_i$ is given by

$$
\hat{\boldsymbol{v}}_i = \sum_{j=1}^{N} \text{softmax}\left(\frac{\boldsymbol{q}_i^\top \boldsymbol{k}_j}{\sqrt{D}}\right)\boldsymbol{v}_j := \sum_{j=1}^{N} a_{ij}\boldsymbol{v}_j.
\tag{4}
$$

The matrix $\mathbf{A} \in \mathbb{R}^{N \times N}$ and its component $a_{ij}$ for $i, j = 1, \cdots, N$ are the attention matrix and attention scores, respectively. Eqn. 3 is also called the "scaled dot-product attention" or "softmax attention". The attention matrix $\mathbf{A}$ after training captures contextual representation for each token.

### 1.1.1 Multi-head Attention

Eqn. 3 corresponds to an attention head. In multi-head attention, output sequences $\hat{\boldsymbol{V}}_h$, $h = 1, \dots, H$ are computed from $H$ attention heads and then concatenated. Let $W^O \in \mathbb{R}^{HD \times HD}$ be the projection matrix for the output. The multi-head attention is defined as

$$
\hat{\boldsymbol{Z}} = \text{MultiHead}(\{\hat{\boldsymbol{V}}\}_{h=1}^H) = \text{Concatenate}(\hat{\boldsymbol{V}}_1, \dots, \hat{\boldsymbol{V}}_H)\mathbf{W}^O.
\tag{5}
$$

The final output of the self-attention layer $T_\ell(\cdot)$ is computed via the following residual connection.

$$
T_\ell(\boldsymbol{X}) = f_\ell(\hat{\boldsymbol{Z}} + \boldsymbol{X}),
\tag{6}
$$

where $f_\ell(\cdot)$ is a function that transforms each feature vector independently and usually chosen to be a feedforward network. In this paper, we call a transformer built with softmax attention softmax transformer or transformer. It is easy to see that both memory and computational complexity of Eqn. 3 are $\mathcal{O}(N^2)$ with $N$ being the length of the input sequence. We can further introduce causal masking into Eqn. 3 for autoregressive applications Vaswani et al. (2017).

Despite the success of transformers in capturing the contextual representation of tokens in the input sequence, it has been shown that the contextual representation learned by the self-attention are redundant. Particularly, attention heads in transformers tend to learn similar attention patterns. Also, many attention scores and keys within each head explain the same patterns and are not needed Michel et al. (2019); Voita et al. (2019b); Bhojanapalli et al. (2021). Such redundancy wastes memory and computation during both training and inference while limiting the model's capacity, posing a challenge to scale up transformers to large-scale tasks.

## 1.2 Contribution

We propose a novel probabilistic model for self-attention, namely the Finite Admixture of Keys (FiAK), that allows pruning attention scores and keys using the prior distributions of attention keys. FiAK models the query distribution $p(\boldsymbol{q}_i)$ as an admixture of Gaussian distributions $\mathcal{N}(\boldsymbol{q}_i \mid \boldsymbol{k}_j, \sigma_j^2 \mathbf{I})$ centering around the attention keys $\boldsymbol{k}_j$, $i, j = 1, \ldots, N$. Our admixture approach uses different mixture models to represent the queries $\boldsymbol{q}_i$ and thus helps increase the diversity of attention patterns. Also, since these mixture models share the same set of component distributions $\mathcal{N}(\boldsymbol{q}_i \mid \boldsymbol{k}_j, \sigma_j^2 \mathbf{I})$, FiAK is efficient. The prior distributions of attention keys in FiAK are then used to prune redundant attention scores and keys to improve the memory and computational cost of the model. An illustration of FiAK and our pruning scheme is given in Fig. 1. Our contribution is three-fold:

1. We develop FiAK, a new finite admixture of keys for self-attention that allows key sharing to diversify attention patterns while guaranteeing the efficiency of the model.

2. We design a probabilistic framework for pruning transformers that employs the prior distributions of keys in FiAK to remove redundant attention scores and keys.

3. We demonstrate the advantages of our FiAK-based pruning protocols on Imagenet object classification, COCO object detection, and WikiText-103 language modeling tasks.

## 1.3 Organization

We structure this paper as follows: In Section 2, we review the connection between attention scores and posterior distributions from a Gaussian mixture model and then present our new finite admixture of keys (FiAK). We formulate our probabilistic pruning framework for transformers via FiAK in Section 3. In Section 4, we validate the advantages of our FiAK-based pruning methods on different benchmarks. In Section 5, we perform empirical analysis of our pruning methods. We discuss related works in Section 6. The paper ends up with concluding remarks. More results and details are provided in the Appendix.

## 2 A Finite Admixture of Keys

In this section, we first review the connection between attention scores in self-attention with the posterior distributions from a Gaussian mixture model (GMM) in Nguyen et al. (2022). We then extend this GMM into a finite admixture of keys (FiAK) to capture more diverse patterns of attention and for pruning attention scores and keys later.

### 2.1 Background: Attention Scores are Posterior Distributions from a Gaussian Mixture Model

Given a query $\boldsymbol{q}_i \in \mathbf{Q}$ and a key $\boldsymbol{k}_j \in \mathbf{K}$, let $\boldsymbol{t}$ be a $K$-dimensional binary random variable having a 1-of-$K$ representation in which a particular element $\boldsymbol{t}_j$ is equal to 1 and all other elements are equal to 0. The distribution $p(\boldsymbol{q}_i|\boldsymbol{t}_j = 1)$ is the likelihood of the query $\boldsymbol{q}_i$ belongs to the $j$-th cluster centering around the key $\boldsymbol{k}_j$. In particular, let $\mathbf{1}$ be an identity matrix and $\pi_j$ be the prior distribution $p(\boldsymbol{t}_j = 1)$, the distribution $p(\boldsymbol{q}_i)$

is given by the following GMM:

$$p(\boldsymbol{q}_i) = \sum_{j=1}^{N} \pi_j p(\boldsymbol{q}_i | \boldsymbol{t}_j = 1) = \sum_{j=1}^{N} \pi_j \mathcal{N}(\boldsymbol{q}_i \,|\, \boldsymbol{k}_j, \sigma_j^2 \mathbf{1}), \tag{7}$$

Following Eqn. 7, the posterior $p(\boldsymbol{t}_j = 1|\boldsymbol{q}_i)$ captures how much the query $\boldsymbol{q}_i$ matches the key $\boldsymbol{k}_j$ and is computed by

$$
\begin{aligned}
p(\boldsymbol{t}_j = 1|\boldsymbol{q}_i) &= \frac{\pi_j \mathcal{N}(\boldsymbol{q}_i \,|\, \boldsymbol{k}_j, \sigma_j^2)}{\sum_{j'} \pi_{j'} \mathcal{N}(\boldsymbol{q}_i \,|\, \boldsymbol{k}_{j'}, \sigma_{j'}^2)} = \frac{\pi_j \exp\left(-\|\boldsymbol{q}_i - \boldsymbol{k}_j\|^2 / 2\sigma_j^2\right)}{\sum_{j'} \pi_{j'} \exp\left(-\|\boldsymbol{q}_i - \boldsymbol{k}_{j'}\|^2 / 2\sigma_{j'}^2\right)} \\
&= \frac{\pi_j \exp\left[-\left(\|\boldsymbol{q}_i\|^2 + \|\boldsymbol{k}_j\|^2\right)/2\sigma_j^2\right] \exp\left(\boldsymbol{q}_i^\top \boldsymbol{k}_j / \sigma_j^2\right)}{\sum_{j'} \pi_{j'} \exp\left[-\left(\|\boldsymbol{q}_i\|^2 + \|\boldsymbol{k}_{j'}\|^2\right)/2\sigma_{j'}^2\right] \exp\left(\boldsymbol{q}_i^\top \boldsymbol{k}_{j'} / \sigma_{j'}^2\right)}.
\end{aligned}
\tag{8}
$$

Assuming that the query $\boldsymbol{q}_i$ and the key $\boldsymbol{k}_j$ are normalized, the prior $\pi_j$ is uniform, and let $\sigma_j^2 = \sigma^2$, $j = 1, 2, \ldots, K$, the posterior $p(\boldsymbol{t}_j = 1|\boldsymbol{q}_i)$ can then be written in the following form

$$p(\boldsymbol{t}_j = 1|\boldsymbol{q}_i) = \frac{\exp\left(\boldsymbol{q}_i^\top \boldsymbol{k}_j / \sigma^2\right)}{\sum_{j'} \exp\left(\boldsymbol{q}_i^\top \boldsymbol{k}_{j'} / \sigma^2\right)} = \mathrm{softmax}\left(\boldsymbol{q}_i^\top \boldsymbol{k}_j / \sigma^2\right). \tag{9}$$

Eqn. (9) becomes Eqn. (4) of the attention score $a_{ij}$ when $\sigma^2 = \sqrt{D}$. Thus, under right assumptions, the attention score $a_{ij}$ between the query $\boldsymbol{q}_i$ and the key $\boldsymbol{k}_j$ in a self-attention layer of a transformer plays the role of the posterior distribution $p(\boldsymbol{t}_j = 1|\boldsymbol{q}_i)$.

### 2.2 FiAK: A Finite Admixture of Keys

The GMM in Eqn. 7 does not take into account the temporal order of the queries $\boldsymbol{q}_i$, i.e. all queries $\boldsymbol{q}_i$ are from the same mixture distribution. In many practical settings such as in the autoregressive tasks, the temporal order of the queries $\boldsymbol{q}_i$ plays an important role. We extend the GMM of keys for self-attention in Eqn. 7 into a finite admixture of keys so that the attention score $a_{ij}$ can capture more diverse attention patterns and provide a probabilistic framework for pruning transformers.

#### 2.2.1 Finite Admixture Models

We first review the finite mixture models (FMMs), such as the GMM in Eqn. 7 above, which served as a workhorse in stochastic modeling, and then discuss the finite admixture models (FAM). A finite mixture distribution of $N$ components for a random array $\mathbf{X} \in \mathbb{R}^{M \times D}$ is given by

$$\boldsymbol{x}_i \sim \sum_{j=1}^{N} p_j f(\boldsymbol{x}; \theta_j), \;\; \sum_{j=1}^{N} p_j = 1, \;\; p_j \geq 0, \tag{10}$$

where $\boldsymbol{x}_i \in \mathbb{R}^D$ is the $i$-th row of $\mathbf{X}$ randomly sampled from the mixture distribution. $f$ is a chosen probability measure, such as a Gaussian distribution as in Eqn. 7, $p = \{p_1, \ldots, p_N\}$ are mixture weights that correspond to the prior $\pi_j$, and $\theta_j$ denotes the parameter values for the $k$-th component.

A FAM is a generalization of a FMM, in which rows $\boldsymbol{x}_i$, $i = 1, \ldots, M$, are drawn from different mixture distributions that share $N$ components $f(\boldsymbol{x}; \theta_j)$, $j = 1, \ldots, N$ with different mixture weights

$$\boldsymbol{x}_i \sim \sum_{j=1}^{N} p_{ij} f(\boldsymbol{x}; \theta_j), \;\; \sum_{j=1}^{N} p_{ij} = 1, \;\; p_{ij} \geq 0. \tag{11}$$

Comparing to FMM, FAM has better representation capacity thanks to its flexibility in choosing the mixture components. Since all components are shared between mixtures in FAM, FAM is efficient in term of the model size and computational cost for sampling samples from the model.

### 2.2.2 Finite Admixture of Keys

We propose the finite admixture of keys (FiAK) for the queries in self-attention. In Eqn. 11, let the function $f(\boldsymbol{x}; \theta_j) = p(\boldsymbol{q}_i | \boldsymbol{t}_j = 1) = \mathcal{N}(\boldsymbol{q}_i | \boldsymbol{k}_j, \sigma_j^2 \mathbf{I})$ and $p_{ij} = \pi_{ij} = p_i(\boldsymbol{t}_j = 1)$ where $\pi_{ij} = p_i(\boldsymbol{t}_j = 1)$ is the prior distribution $p(\boldsymbol{t}_j = 1)$ of the mixture corresponding to the query $\boldsymbol{q}_i$. FiAK is defined as follows:

**Definition 1** (Finite Admixture of Keys)**.** *Given a set of queries $\boldsymbol{q}_i$ and keys $\boldsymbol{k}_j$ in self-attention, $i, j = 1, \ldots, N$, the queries $\boldsymbol{q}_i$ admit a finite admixture of keys if $\boldsymbol{q}_i$ are sampled from the following finite admixture model:*

$$\boldsymbol{q}_i \sim \sum_{j=1}^{N} \pi_{ij} p(\boldsymbol{q}_i | \boldsymbol{t}_j = 1) = \sum_{j=1}^{N} \pi_{ij} \mathcal{N}(\boldsymbol{q}_i | \boldsymbol{k}_j, \sigma_j^2 \mathbf{I}), \ \sum_{j=1}^{N} \pi_{ij} = 1, \ \pi_{ij} \geq 0. \tag{12}$$

Note that the difference between FiAK and the Gaussian mixture of keys in Eqn. 7 is that the prior distribution $\pi_{ij}$ are different for each query $\boldsymbol{q}_i$, i.e. $\boldsymbol{q}_i$ are sampled from different mixtures that share the same $N$ components $\mathcal{N}(\boldsymbol{q}_i | \boldsymbol{k}_j, \sigma_j^2 \mathbf{I})$, $i, j = 1, \ldots, N$.

**Remark 1** (FiAK as a Topic Model)**.** *FiAK can be connected to the Probabilistic Latent Semantic Analysis (pLSA) model for topic modeling Hofmann (1999). Given the document d that contains the word w sampled from the topic c, pLSA models the occurrence of the word w in the document d as a mixture of conditionally independent Multinomial distributions $p(w|d) = \sum_c p(c|d)p(w|c)$. Comparing this equation of pLSA and Eqn. 12 of FiAK, we can interpret the mixture weights $\pi_{ij}$ and the distribution $\mathcal{N}(\boldsymbol{q}_i | \boldsymbol{k}_j, \sigma_j^2 \mathbf{I})$ in FiAK as the distributions $p(c|d)$ and $p(w|c)$ in pLSA, respectively. As a result, the attention keys in FiAK correspond to the topics, and the queries are words sampled from those topics. pLSA and thus FiAK are also equivalent to the well-known Latent Dirichlet Allocation model under a uniform Dirichlet prior on the per-document topic distribution $p(c|d)$ Blei et al. (2003).*

## 3 Prior-based Pruning via FiAK

Using the prior $\pi_{ij}$ in FiAK, we propose two novel pruning methods: 1) attention score pruning via FiAK and 2) mixed pruning via FiAK. For comparison with the GMM of keys in Section 2.1, we also derive 3) key pruning via GMM. Here, attention score pruning removes the redundant attention scores $a_{ij}$, key pruning removes the redundant attention keys $\boldsymbol{k}_j$ together with its corresponding value vectors $\boldsymbol{v}_j$, and mixed pruning removes both attention scores and keys, as well the corresponding value vectors $\boldsymbol{v}_j$ as in key pruning. In all of our proposed methods, attention scores and keys with the smallest importance weights, i.e. $|\hat{\pi}_{ij}|$, $\hat{S}(j)$, and $|\hat{\pi}_j|$ in Algorithm 1, 2, and 3 are pruned away.

**Attention Score Pruning via FiAK.** The magnitude of the prior, $|\pi_{ij}|$, in FiAK implies how much the key $\boldsymbol{k}_j$ is needed to explain the query $\boldsymbol{q}_i$. These priors act as importance weights of the keys $\boldsymbol{k}_j$ given the query $\boldsymbol{q}_i$ and can be used to prune away the attention score $a_{ij} = \text{softmax}\left(\frac{\boldsymbol{q}_i^\top \boldsymbol{k}_j}{\sqrt{D}}\right)$, thus saving memory and computation when computing the self-attention (see Algorithm 1).

**Mixed Pruning via FiAK.** To further reduce the computation complexity of the model, we introduce mixed pruning via FiAK in Algorithm 2. In addition to pruning the attention score $a_{ij}$, we derive the importance weights of the keys $\boldsymbol{k}_j$ and remove the pairs $(\boldsymbol{k}_j, \boldsymbol{v}_j)$ whose importance weights are the smallest. This strategy enables the pruned model to save computation not only at the attention calculation step, but also complete removes the key vector $\boldsymbol{k}_j$ and the value vector $\boldsymbol{v}_j$, as well as other computations related to these vectors in Eqn. 4.

For autoregressive tasks like language modeling, the attention matrices are lower triangular which makes the importance scores $\hat{S}(j) = \sum_i |\hat{\pi}_{ij}|$ in Algorithm 2 become biased for different time steps $j$ of the keys. In particular, the smaller $j$, the more $|\hat{\pi}_{ij}|$ are added into the sum since $|\hat{\pi}_{ij}| = 0$ for $i < j$, $i, j = 1, \ldots, N$, i.e. a query can only attend to the current and previous keys, but not the future keys. To address this problem, we normalize the sum by the number of the entries below the diagonal for each column when computing the importance score as in Step 3 of Algorithm 2.

---

**Algorithm 1** Attention Score Pruning via FiAK

---
**Hyperparameter** $0 < k < 1$: $k$ fraction of the attention scores $a_{ij}$ to be pruned.
**Step 1** Incorporate parameters $\pi_{ij}$ into the self-attentions.
**Step 2** Train the transformer with the additional parameters $\pi_{ij}$ until convergence.
**Step 3** Prune $k$ fraction of the attention scores $a_{ij}$ whose learned coefficients $|\hat{\pi}_{ij}|$ are the smallest.
**Step 4** Set the remaining $\hat{\pi}_{ij} = 1$, which corresponds to uniform prior, and finetune the pruned network.

---

**Algorithm 2** Mixed Pruning via FiAK

---
**Hyperparameters** $0 < k_1, k_2 < 1$: $k_1$ fraction of the total attention scores $a_{ij}$ to be pruned; $k_2$ fraction of pairs (key, value) to be pruned.
**Step 1** and **Step 2** Same as **Step 1** and **Step 2** of Algorithm 1.
**Step 3** Calculate the importance score $\hat{S}(j)$ of each pair $(\boldsymbol{k}_j, \boldsymbol{v}_j)$:

$$\hat{S}(j) = \sum_i |\hat{\pi}_{ij}|, \text{ or } \hat{S}(j) = \frac{1}{N-j+1}\sum_i |\hat{\pi}_{ij}| \text{ if the task is autoregressive.}$$

Then prune $k_2$ fraction of the pairs $(\boldsymbol{k}_j, \boldsymbol{v}_j)$ with the smallest scores $\hat{S}(j)$.
**Step 4** Prune $\hat{k}_1$ fraction of the remain unpruned $a_{ij}$ whose corresponding $|\hat{\pi}_{ij}|$ are the smallest $\hat{k}_1 = 1 - \frac{1-k_1}{1-k_2}$.
**Step 5** Follow **Step 4** of Algorithm 1.

---

**Algorithm 3** Key Pruning via GMM

---
**Hyperparameter** $0 < k < 1$: $k$ fraction of the keys to be pruned.
**Step 1** Incorporate parameters $\pi_j$ into the self-attentions.
**Step 2** Train the transformer with the additional parameters $\pi_j$ until convergence.
**Step 3** Prune $k$ fraction of the key-value pairs $(\mathbf{k_j}, \mathbf{v_j})$, whose corresponding learned mixing-coefficients $|\hat{\pi}_j|$ are the smallest.
**Step 4** Set the remaining $\hat{\pi}_j = 1$, i.e. uniform prior, and finetune the pruned network.

---

Table 1: Computational saving achieved by using our proposed pruning methods to prun a dense H-head softmax attention. $H$, $N$, $D$, $D_x$ denote the number of attention heads, sequence length, head dimension, and model/input dimension, respectively. Parameters $k$ and $(k_1, k_2)$ are the fraction to be pruned as explained in Algorithm 1, 2 and 3. Advantages of our pruned models increase for larger models and longer sequences.

| Method | Hyper-parameters | Computational Saving for H-head Attention |
|---|---|---|
| Attention Score Pruning via FiAK | $k$ | $kHN^2(2D-1)$ |
| Mixed-Pruning via FiAK | $k_1, k_2$ | $2[(k_1+k_2)D - k_1]HN^2 + (2D_x - 3)k_2 HDN$ |
| Key Pruning via GMM | $k$ | $kHN^2(4D-1) + (2D_x - 3)kHDN$ |

**Key Pruning via Gaussian Mixture Model.** For a comparison between admixture-based pruning and mixture-based pruning, we introduce key pruning via GMM (Algorithm 3), which uses the learned prior $|\pi_j|$ in the GMM defined by Eqn. 7 as importance weights to prune the pairs $(\boldsymbol{k}_j, \boldsymbol{v}_j)$.

**Finetuning the Pruned Network.** FiAK (Eqn.12) introduces additional priors $\pi_{ij}$ to capture the importance of the attention score $a_{ij}$. After the attention scores are pruned, those extra parameters can be removed by setting them to 1, which corresponds to using uniform priors. The network is then finetuned for more epochs to obtain competitive accuracy compared to the dense baseline network. The same finetuning strategy is used for key pruning via GMM.

**Computational Complexity Reduction from Pruning.** Table 1 shows the computational saving from the multi-head attention pruned by our methods compared to the dense baseline softmax attention. Here, H is the number of attention heads, and the computational cost is computed as the total number of additions and multiplications needed. Our analysis results in Table 1 indicate that our pruning methods save more computations as we scale up the model for longer sequences, i.e. the sequence length $N$ is large. Note that the saving in computational cost is quadratic in terms of $N$. Our derivation are given in Appendix D.

Table 2: Top-1 and top-5 accuracy (%) of the pruned models from the attention score and mixed pruning via FiAK on the Imagenet dataset compared to the dense baseline DeiT-tiny Touvron et al. (2020a). We also show the top-1 and top-5 accuracy (%) of the pruned model from the key pruning via GMM on the same task for comparison. Here, pruning fractions of attention scores via FiAK and keys via GMM are $k$. For mixed pruning via FiAK, pruning fractions for attention scores and keys are $k_1$ and $k_2$, respectively. FiAK-based pruning schemes result in pruned models with much better accuracies than the dense baseline and those from the GMM-based pruning scheme.

| Method | Top-1 Acc | Top-5 Acc |
|---|---|---|
| *Baseline DeiT-tiny* | 72.23 | 91.13 |
| GMMformer | 72.96 | 91.64 |
| Key pruned GMMformer $k = 30\%$ | 71.57 | 90.80 |
| FiAKformer | 73.50 | 91.90 |
| Attention-score pruned FiAKformer $k = 50\%$ | 73.56 | **91.95** |
| Attention-score pruned FiAKformer $k = 60\%$ | **73.67** | 91.91 |
| Attention-score pruned FiAKformer $k = 70\%$ | 73.09 | 91.57 |
| Mixed pruning FiAKformer $k_1 = 70\%$, $k_2 = 15\%$ | 72.78 | 91.38 |
| Mixed pruning FiAKformer $k_1 = 70\%$, $k_2 = 20\%$ | 72.25 | 91.14 |

## 4 Experimental Results

We empirically corroborate the advantages of the models pruned via our proposed FiAK-based pruning methods over the dense baseline model on various benchmarks, including ImageNet object classification, COCO object detection, and WikiText-103 language modeling. We aim to show that: (i) the FiAK-based pruned models are more efficient than the dense baseline in term of memory and computation cost while achieving comparable/better accuracy; (ii) the FiAK-based pruning methods, i.e. attention score pruning and mixed pruning via FiAK, are more effective than the GMM-based pruning method, i.e. key pruning via GMM, resulting in more efficient and accurate pruned models.

Throughout this section, we refer to tranformers that use FiAK-based attention defined by Eqn. 12 as FiAKformer and transformers that use GMM-based attention defined by Eqn. 7 as GMMformer. Except for the ImageNet object classification task, in other experiments in this section, we use the attention score pruning via FiAK to study the FiAK-based pruning. The details on datasets, models, and training are provided in Appendix A.

### 4.1 Image classification on Imagenet

**Model and setting.** We use the DeiT-tiny model Touvron et al. (2020a) with 12 transformer layers and 4 attention heads per layer. The model dimension is 192. To train the models, we follow the same setting and configuration as for the baseline Touvron et al. (2020a), with the initialization of the learnable priors $\pi_{ij}$ and $\pi_j$ set to be $\frac{1}{\sqrt{N}}$ and $\frac{1}{N}$, respectively, where $N$ is the length of the input sequence.

**Results.** *Pruned models from attention score and mixed pruning via FiAK attain much better accuracy than the DeiT-tiny baseline while being significantly more efficient (See Table 2).* Attention score pruning via FiAK at different pruning fractions $k = 50\%$, 60% and 70% result in the highest accuracies. In particular, at the pruning fractions $k = 50\%$ and 60%, we observe substantial accuracy improvement over the dense baseline (1.33% and 1.44% in top-1 accuracy, respectively). These two pruned models also outperform the dense FiAKformer. On the other hand, mixed pruning with the same attention score pruning fraction, $k_1 = 70\%$ and different key pruning fractions, $k_2 = 15\%$ and 20%, gain better accuracy compared to the baseline while obtaining the most computation and memory reduction (See Fig. 2). These results show the effectiveness of our pruning schemes for the image classification task. The efficiency of our pruned models on this task is discussed in Section 5.1.

Comparison with GMM-based pruning. Table 2 shows that while the GMMformer yields better accuracy than the baseline, by pruning 30% of attention keys (or equivalently key-value pairs), the pruned model performs worse than the baseline. *This results justify the advantage of the FiAK-based pruning over the GMM-based pruning and validate the need of using admixture, such as FiAK, to model the self-attention and design its effective pruning schemes.*

Table 3: Comparison to other pruning methods on Imagenet task.

| Method | FLOPS reduction (%) | Acc-1 (%) |
|---|---|---|
| *DeiT-tiny* | 0.00 | 72.23 |
| Head pruning Michel et al. (2019) | **23.69** | 68.59 |
| $S^2ViTE$ Chen & et al. (2021) | **23.69** | 70.12 |
| Attention-score pruned FiAKformer k = 70% | 8.50 | **73.09** |
| Mixed pruned FiAKformer $k_1 = 70\%$, $k_2 = 20\%$ | 13.00 | 72.25 |
| Mixed pruned FiAKformer $k_1 = 70\%$, $k_2 = 20\%$ + $S^2ViTE$ Chen & et al. (2021) | 22.76 | 72.24 |

Table 4: Box mean average precision (box mAP) on the COCO validation set of the Swin Tranformer Liu et al. (2021) pruned with mixed pruning via FiAK in Algorithm 2 compared to the baseline dense Swin transformer. The mixed pruned model with the pruning fraction $k_1 = 70\%$ and $k_2 = 15\%$ achieves a comparable result with the dense baseline while outperforming the attention-score pruned model that prunes the model less with a smaller pruning fraction $k = 60\%$. The box mAP of the pretrained baseline is reported at https://github.com/SwinTransformer/Swin-Transformer-Object-Detection.

| Method | Box mAP |
|---|---|
| *Baseline Swin transformer* | 43.7 |
| FiAKformer | **44.7** |
| Attention-score pruned FiAKformer 60% | 43.3 |
| Mixed pruned FiAKformer $k_1 = 70\%$, $k_2 = 15\%$ | 43.6 |

**Comparison to Other Pruning Methods.** We compare our FiAK-based pruning schemes with other pruning methods for transformers on ImageNet task (see Table 3 below). Compared to the head pruning Michel et al. (2019) and $S^2ViTE$ Chen & et al. (2021), our schemes prune the model less but increase its accuracy. Combining with the $S^2ViTE$ Chen & et al. (2021), mixed FiAK pruning can increase the FLOPs reduction up to 22.76% while maintaining similar advantage in accuracy on the ImageNet task.

## 4.2 Object Detection on COCO with a Pre-trained Model.

Pruning methods via FiAK are universal and can also be applied on pre-trained transformers finetuned for downstream tasks. Here, we demonstrate the effectiveness our FiAK-based pruning schemes for a pre-trained Swin transformer finetuned for the object detection task on the COCO dataset Lin et al. (2014).

**Model and baselines** *Pruning via FiAK methods are easy to apply on the pre-trained models.* Following Algorithm 1, we apply the attention score pruning via FiAK on the pre-trained tiny Swin Transformer Liu et al. (2021) for the object detection task on the COCO dataset. We first introduce additional parameters for the priors $\pi_{ij}$ into self-attentions of the pre-trained model and then learn these priors by finetuning the model for 12 epochs. After pruning the attention scores $a_{ij}$, we do another finetuning round with uniform priors as in Algorithm 1. The entire process takes less than one-tenth of the time used for training a Swin transformer from scratch, indicating the efficiency of applying our method. We follow the same configuration and setting for the baseline as provided in Liu et al. (2021).

**Results** At pruning fraction $k = 60\%$, the Swin transformer pruned with attention score pruning via FiAK obtains the Box mean average precision (box mAP) of 43.3, which is comparable with the box mAP 43.7 of the dense baseline. Note that the dense Swin-FiAKformer attains the box mAP = 44.7, which significantly better than the box mAP of the dense basline. The mixed pruned model via FiAK with pruning fractions $k_1 = 70\%$ and $k_2 = 15\%$ has comparable performance with the baseline dense softmax model ( mAP = 43.6 vs. mAP = 43.7). It is interesting to notice that the mixed pruned model also outperforms the attention-score pruned model while allowing larger pruning fraction, 70% vs. 60%. Also, mixed pruned models have smaller computational complexity than attention-score pruned models as explained in Section 3 and Table 1. The simple usage and competitive performance of pruning via FiAK on this task demonstrate the universal applicability of our methods.

Table 5: Test perplexity of pruned FiAKformer for the language modeling task on Wikitext-103 dataset. We apply attention score pruning via FiAK and prune 40% and 50% of the attention scores. We also apply mixed pruning via FiAK with $k_1 = 40\%$ and $k_2 = 10\%$. The results show that the pruned models after finetuning can reach comparable or better accuracy than the dense baseline.

| Method | Perplexity (PPL) |
| --- | --- |
| *Baseline softmax transformer* | 34.29 |
| FiAKformer | **33.69** |
| Attention score pruned FiAKformer 40% | 33.88 |
| Attention score pruned FiAKformer 50% | 34.28 |
| Mixed pruning FiAKformer $k_1 = 40\%$, $k_2 = 10\%$ | 34.21 |

### 4.3 Language Modeling on WikiText-103

To examine the effectiveness of our pruning methods across different data modalities, we experiment with the word-level language modeling task on WikiText-103 Merity et al. (2017). For this autoregressive task, we use the normalization procedure for mixed pruning via FiAK described in Algorithm 2.

**Models and baselines** The baseline model we use in our experiments is the 8-head softmax baseline transformer with 16 layers. We follow the experiment settings from Schlag et al. (2021).

**Results** We summarize our results in Table 5. Same as the vision tasks above, attention score pruning via FiAK and mixed pruning via FiAK yield more efficient language models with competitive or even better performance than the dense baseline.

## 5 Empirical Analysis

### 5.1 Efficiency Analysis

We investigate the improvement in efficiency of transformers pruned via FiAK-based and GMM-based approach over the baseline. In particular, we analyze the computation and memory complexity of the pruned models trained for the ImageNet object classification task in Section 4.1. For attention score pruning via FiAK (Algorithm 1), we study the pruned model with pruning fraction $k = 70\%$. For mixed pruning via FiAK (Algorithm 2), we study the pruned model with pruning fractions $k_1 = 70\%$ and $k_2 = 20\%$. For comparison, we also analyze key pruning via GMM (Algorithm 3) using a pruned model with pruning fraction $k = 30\%$. As shown in Table 2, model pruned by key pruning via GMM with pruning fraction $k = 30\%$ already yields worse accuracy than models pruned via FiAK. Additional efficiency analysis for our FiAK-based pruning methods on the WikiText-103 language modeling task is provided in Appendix C.

*Efficiency Advantage of Models Pruned via FiAK over the Baseline Model Grows with the Sequence Length.* In Fig. 2, we compare the floating point operations per second (FLOPS) ratios and the memory ratios at inference time between the models pruned via FiAK and the dense baseline for multiple sequence lengths {197, 785, 3137}. In Fig. 2, (A) and (C) shows the FLOPS and memory ratios for the attention blocks, respectively, while (B) and (D) shows these ratios for the whole model, respectively. In (C) and (D), the memory ratios for attention score and mixed pruning via FiAK are the same. Overall, we observe that our pruning schemes result in much more efficient models for long sequences in terms of computation and memory compared to the dense baseline, and this advantage becomes more significant for longer sequences.

*FiAK-based pruning also wins in real time.* On ImageNet task, the latency for the dense baseline and our attention-score pruned FiAKformer, $k = 70\%$, are 508 and 649 images/second (on GPU) and 76 and 95 images/second (on CPU), respectively.

*Mixed-pruning via FiAK gains the most advantage in FLOPS among different pruning methods while still achieving competitive performance to the dense baseline.* This trend continues as the sequence length increases. Attention score and mixed pruning via FiAK share the same benefits of substantial memory reduction. As shown in Fig. 2(C) and (D), at sequence length $N = 785$ and 3137, the pruned models via FiAK save more than 60% of the baseline memory, both for each multi-head attention block and the whole model.

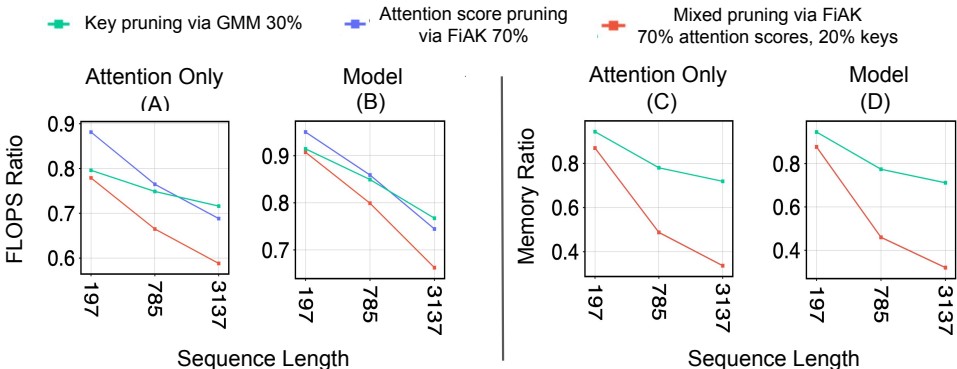

Figure 2: FLOPS and memory ratios at inference between the models pruned with FiAK-based/GMM-based pruning schemes and the dense Deit-tiny baseline. For a thourough analysis, we show a comparison at attention block only ((A) and (C)) and for the entire model ((B) and (D)). Note that in (C) and (D), the memory ratios for attention score and mixed pruning via FiAK are the same. The advantage of the FiAK-based pruning grows with the sequence length. Compared to the GMM-based pruning, FiAK-based pruning schemes are more effective.

*FIAK-based pruning methods result in more efficient models with better accuracy than the GMM-based pruning* (See Table 2). This again proves the advantage of modeling self-attention as an admixture model rather than a mixture model.

## 5.2 Visualizing the pruning masks

Figure 3 shows the pruning masks for the model trained on the ImageNet object classification task in Section 4.1. In particular, we visualize the magnitude of the priors $\pi_{ij}$ at each of the 4 heads in layers {1, 3, 5, 7, 9, 11} (Left), as well as the binary pruning masks when we prune 70% of the attention scores with attention score pruning via FiAK (Right). We observe that the matrices of the priors $\pi_{ij}$ (Left) are sparse with large values of $\pi_{ij}$ are near the diagonal. This suggests that the queries $q_i$ pay attention to the keys $k_j$ in its local neighborhood. Therefore, when pruning is applied, we can remove redundant attention scores $a_{ij}$ where $|i - j|$ is large. Also, Figure 4 visualizes the pruning masks obtained with attention score pruning via FiAK for the WikiText-103 language modeling task. Additional results on visualizing the pruning masks obtained with mixed pruning is provided in Appendix B.

We observe the differences between the masks for different attention heads. We hypothesize that the FiAK-based pruning schemes learn different pruning masks between heads (see Figures 2, 4, and 5) in order to diversify the attention patterns and reduce the well-known head redundancy. We have confirmed this hypothesis by computing the average $\mathcal{L}_2$ distance between attention heads in each layer of the models pruned with the attention score pruning via FiAK and of the baseline dense model for the WikiText-103 language modeling task. We observe that the layer-average mean of these distances in the attention-score pruned FiAKformer with $k = 40\%$ ($7.22 \pm 2.11$) is greater than in the baseline dense model ($6.20 \pm 2.30$), suggesting that the FiAK-based pruned model learns more diverse attention patterns.

In addition, our FiAK-based pruning learns different masks at different layers as shown in Figures 2, 4, and 5. For the ImageNet task (see Figures 2 and 5), early layers have small receptive fields that capture local and low-level features. Thus, the masks learned by the FiAK-based pruning at these early layers have local patterns centered around the main diagonals. Later layers have large receptive fields that capture global and high-level features. Therefore, the FiAK-based pruning tends to learn non-local masks at these later layers. The difference between pruning masks learned at different layers for the WikiText-103 language modeling task does not have an interpretable pattern.

## 5.3 Pruning Masks Obtained via FiAK are Training-Agnostic

In this section, we show that the pruning masks learned via FiAK can still capture positions of important attention scores in the baseline model trained without additional prior parameters $\pi_{ij}$ in Eqn. 12. In our experiments on the ImageNet object classification task, we mask the attention matrices in the pretrained DeiT-tiny provided in Touvron et al. (2020a) by the binary pruning masks obtained from the attention score

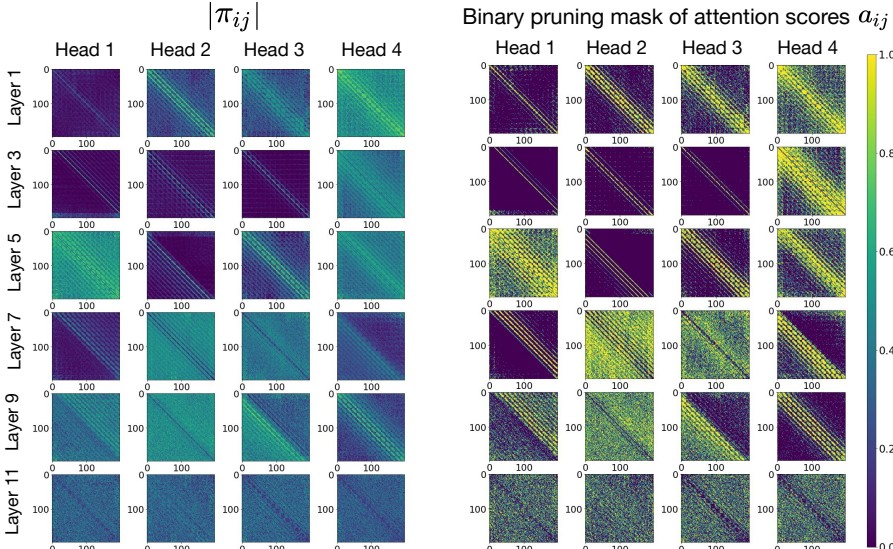

Figure 3: (Left) The magnitude of the priors, $|\pi_{ij}|$, learned from the ImageNet object classification task, and (Right) the binary pruning masks of the attention scores for attention score pruning via FiAK with the pruning fraction $k = 70\%$. We plot these prior matrices and pruning masks for all 4 heads in layer $\{1, 3, 5, 7, 9, 11\}$. The visualization of $|\pi_{ij}|$ shows that the prior matrices are sparse, allowing most attention scores to be pruned.

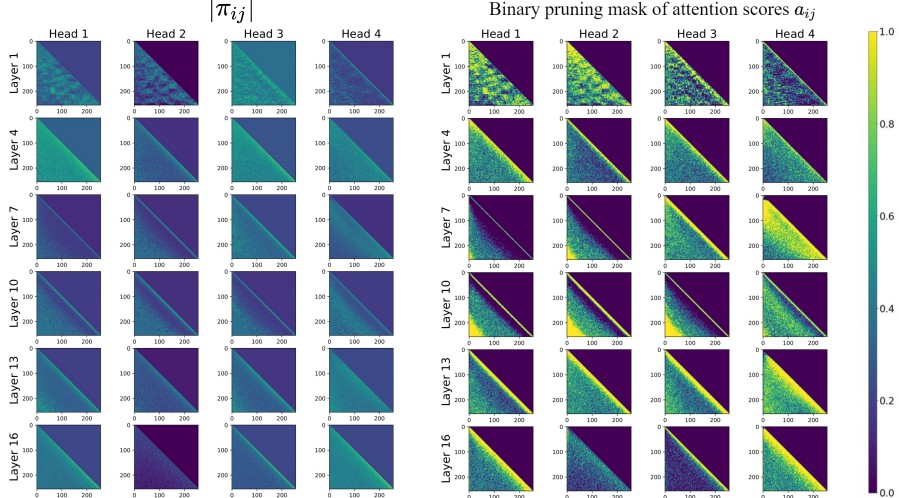

Figure 4: (Left) Magnitude of the priors $|\pi_{ij}|$ learned from the language modeling task. (Right) The binary pruning masks from attention score pruning via FiAK with the pruning fraction $k = 50\%$. We plot the prior matrices and pruning masks for the first 4 heads in layers $\{1, 4, 7, 10, 13, 16\}$.

pruning via FiAK with the pruning fraction $k = 60\%$. The masked model is then fine-tuned till convergence. Table 6 shows that the masked DeiT-tiny transformer achieves better performance than the dense baseline, demonstrating that the pruning masks learned by FiAK capture relevant attention scores and meaningful connections between tokens at each heads in the baseline DeiT-tiny trained with the setting in Touvron et al. (2020a). This result suggests that the pruning masks learned via FiAK are training-agnostic. After learned, these masks can be applied on the baseline models trained with different training schemes.

## 6    Related Work

**Pruning and Reducing Redundancy in Transformers** It has been shown that most of the neurons and heads in the pre-trained transformer are redundant and can be pruned when applied on a downstream task Dalvi et al. (2020); Michel et al. (2019); Durrani et al. (2020). Works in pruning transformers can be categorized into two groups: 1) head pruning and 2) token pruning. An early work in head pruning calculates

Table 6: Top-1 and top-5 accuracy (%) of the masked vs dense DeiT-tiny Touvron et al. (2020a). We mask the attention matrices of the trained baseline with the binary pruning masks learned from attention score pruning via FiAK with the pruning fraction $k = 60\%$ and then fine-tune the masked model. The masked DeiT-tiny achieves better performance than the dense baseline, indicating that the pruning masks can capture important attention scores.

| Method | Acc Top-1 | Acc Top-5 |
|---|---|---|
| *Baseline softmax transformer* | 72.23 | 91.13 |
| Masked softmax transformer $k = 60\%$ | **72.41** | **91.30** |

the head sensitivity to decide to prun a head or not Michel et al. (2019). Voita et al. (2019a) employs layerwise relevance propagation to decide the head importance. The head importance can also be learned in a data-driven manner as in Li et al. (2021). For token pruning, Goyal et al. (2020) computes a token's importance score as average attention score of other tokens to that token. A dropout-based approach that stochastically determines a sequence length at each layer has also been used to prune redundant tokens Kim & Cho (2021). Kim et al. (2021) proposes an adaptive approach that learns an attention mask for token pruning. The contextualized embeddings in pre-trained networks under this redundancy due to overparameterization have also been studied to demonstrate that the representations learned within these models are highly anisotropic Mu & Viswanath (2018); Ethayarajh (2019). Knowledge distillation and sparse approximation have also been used to enhance the efficiency of transformers, including Sanh et al. (2019); Sun et al. (2019); Voita et al. (2019b); Sajjad et al. (2020). Our FiAK-based approach is complementary to these methods.

**Efficient Transformers** To lower the quadratic computational and memory cost of transformers, efficient transformers have been studied Roy et al. (2021). Among them are sparse transformers which incorporate sparse structures into the attention matrix Parmar et al. (2018); Liu et al. (2018); Qiu et al. (2019); Child et al. (2019); Beltagy et al. (2020). Another class of efficient transformers are models that aim to have better coverage by integrating different access patterns Child et al. (2019); Ho et al. (2019), which can also be learned from the data Kitaev et al. (2020); Roy et al. (2021); Tay et al. (2020). In other works, a side memory module is utilized in order to access multiple tokens simultaneously Lee et al. (2019); Sukhbaatar et al. (2019); Asai & Choi (2020); Beltagy et al. (2020). In another line of work, low-rank and kernelization methods have recently been proposed to improve the computational and memory efficiency of self-attention calculation Tsai et al. (2019); Wang et al. (2020); Katharopoulos et al. (2020); Choromanski et al. (2021); Shen et al. (2021); Nguyen et al. (2021); Peng et al. (2021). Our FiAK is orthogonal to these methods.

**Mixture Models for Transformers** Recently, mixture models have been employed to study and improve transformers. Among these works is the transformer with a mixture of Gaussian keys in Nguyen et al. (2022). This work develops a probabilistic framework underlying attention mechanism in transformers, which we review in Section 2.1, and propose a new transformer with a mixture of Gaussian keys, that replaces redundant heads in transformers with a mixture of keys at each head. (Zhang & Feng, 2021) develops a Gaussian mixture attention that models each attention score as a GMM. Another work is the switch transformers Fedus et al. (2021) that make use of the routing algorithm in Mixture of Experts (MoE) to reduce the communication and computational costs in transformers. Other works that combine mixture models with transformers include Cho et al. (2020); Guo et al. (2019); Jiang et al. (2020).

# 7 Concluding Remarks

In this paper, we propose FiAK, a novel finite admixture of keys for self-attention, that model the distribution of queries $q_i$ in self-attention as an admixture of Gaussian distributions $\mathcal{N}(q_i \mid k_j, \sigma_j^2 \mathbf{I})$ whose centers are the attention keys $k_j$, $i, j = 1, \ldots, N$. Using the prior distributions of the attention keys in FiAK, we propose a probabilistic pruning framework to remove redundant attention scores and keys in transformers. We verify that models pruned by our FiAK-based pruning methods improve the memory and computational cost over the baseline dense transformers while achieving comparable or better accuracy. As mentioned in Remark 1, admixture models are equivalent to Latent Dirichlet Allocation (LDA) models under a uniform Dirichlet prior. Extending FiAK into an LDA-based framework for pruning transformers is an interesting research direction for future work.

**Broader Impact Statement**

Given the nature of the work, we do not foresee any negative societal and ethical impacts of our work.

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

## Appendix for "Pruning Transformers with a Finite Admixture of Keys"

In this appendix, we include experimental details, additional experiments/visualization of pruning via FiAK, and the detailed derivation of the computational saving in Table 1. We also provide code to reproduce our results in a separate folder in the supplementary material.

## A    Experiment details

In this section, we provide model and training details of our experiments in Section 4.

### A.1    Object classification on Imagenet

Our baseline for the object classification task is the DeiT-tiny, a 4-head dense softmax transformer with 12 layers. In this baseline, the model dimension is of size 192, the feed-forward layer is of size 768, and the patch size is 16, which is equivalent to the sequence length of 197. Our FiAKformers and GMMformers have the same architecture configuration as the baseline with additional parameters $\pi_{ij}$ and $\pi_j$ as in Eqn.12 and 7, respectively.

All models are trained for 300 epochs, and the pruned models are fine-tuned on the Imagenet dataset for additional 100 epochs, using 4 A100 GPUs, 40 GB each, with batch size of 256. The initial learning rates for training and fine-tuning are $5 \times 10^{-4}$ and $5 \times 10^{-5}$, respectively.

### A.2    Object Detection on COCO

The pretrained baseline for the object detection task is the Swin-Transformer-tiny provided at https://github.com/SwinTransformer/Swin-Transformer-Object-Detection. All models have 12 attention layers with windows of size 7, which is equivalent to the sequence length of 49 patches per window.

The pruned models are fine-tuned for 12 epochs, using 8 A100 GPUs, 40GB each, with the initial learning rate is $10^{-4}$. The weights are updated by an AdamW optimizer with the weight decay coefficient of 0.05.

### A.3    Language Modeling on WikiText-103

For the language modeling task, we use the dense softmax transformer as our baseline. For all experiments, we use a transformer model that has 16 layers, 8 heads, feed-forward layer dimension of size 2048, embedding dimension of size 128, and hidden dimension of size 128. The context length for training and evaluation is set to 256.

We train our models using 2 A100 GPUs, 40GB each. We set the batch size to 96 and train our models for 120 epochs. We also apply dropout with dropout rate 10%. To optimize our models, we use Adam optimizer and Cosine annealing scheduler with initial learning rate 0.00025.

After the training phase, we prune the resulting models using one of our pruning schemes. Then, we finetune the pruned models for 30% time of the training phase, or equivalently 36 epochs.

## B    Additional Results on Visualizing the Pruning Masks

In this section, we provide additional results on visualizing the pruning masks learned from pruning via FiAK. Figure 5 depicts the pruning mask obtained with mixed pruning via FiAK for the ImageNet object classification task.

In Figure 6, we provide a detailed visual analysis of the parameters $\pi_{ij}$ learned from the ImageNet object classification task. We obtained the query-centered mean values (Left) for each attention head by taking the mean of all $\pi_{ij}$ values corresponding to the relative difference in position between the key and query. This results in the aggregation of the pattern learned by $\pi_{ij}$. We also show detailed patterns for the first 5 queries and keys (Right). Each cell in the $5 \times 5$ grid corresponds to the pruning mask applied to each query $\boldsymbol{q}_i$. From Figure 6, we observe that the pruning masks in early layers capture local/short-range attentions while the pruning masks in later layers capture non-local/long-range attentions.

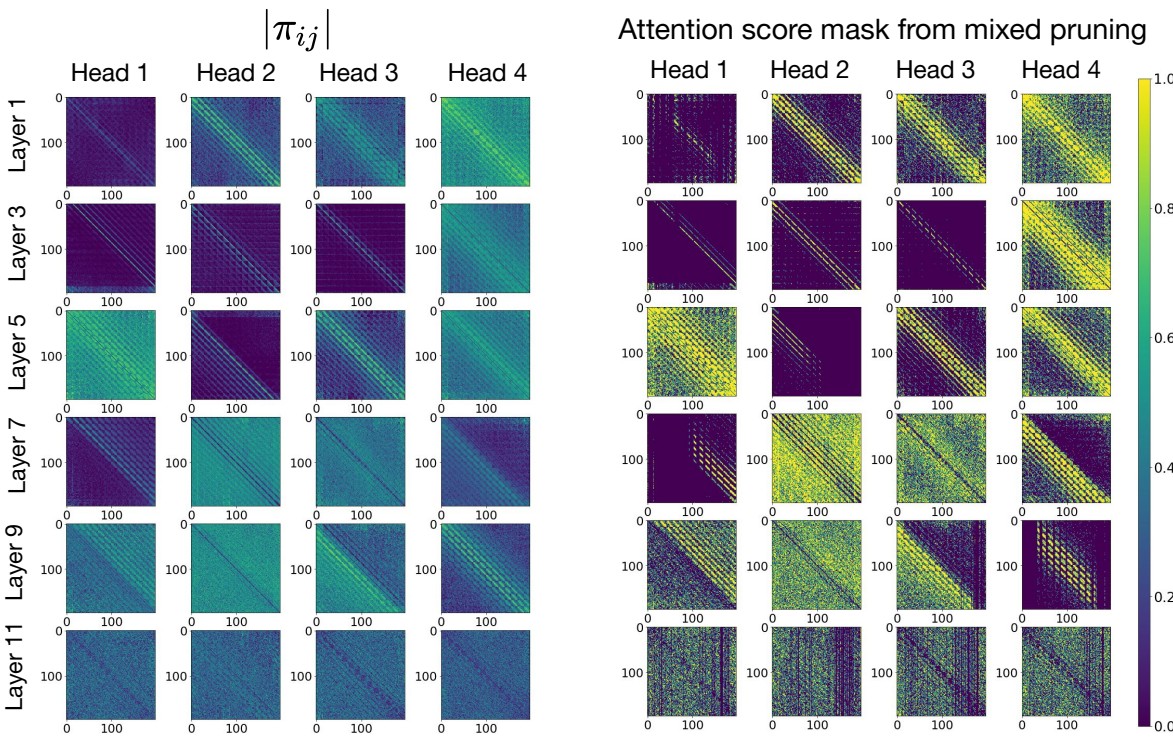

Figure 5: (Left) The magnitude of the priors $|\pi_{ij}|$ learned from the ImageNet object classification task. (Right) The binary attention score masks from mixed pruning via FiAK with the pruning fraction $k_1 = 70\%$ and $k_2 = 15\%$. We plot these prior matrices and pruning masks for all 4 heads in layers $\{1, 3, 5, 7, 9, 11\}$.

.

## C   Additional Efficiency Analysis

In this section, we provide additional efficiency analysis for our pruning methods on the WikiText-103 language modeling task. In particular, we study the pruned models using attention score pruning via FiAK with the pruning fraction $k = 50\%$. We compare the FLOPS and memory ratios between the pruned model and the dense softmax transformer baseline at various sequence lengths $\{128, 256, 512, 1024, 2048, 4096\}$. As shown in Fig. 7, as the sequence length grows, our pruned model becomes significantly more efficient in both memory and computations than the baseline.

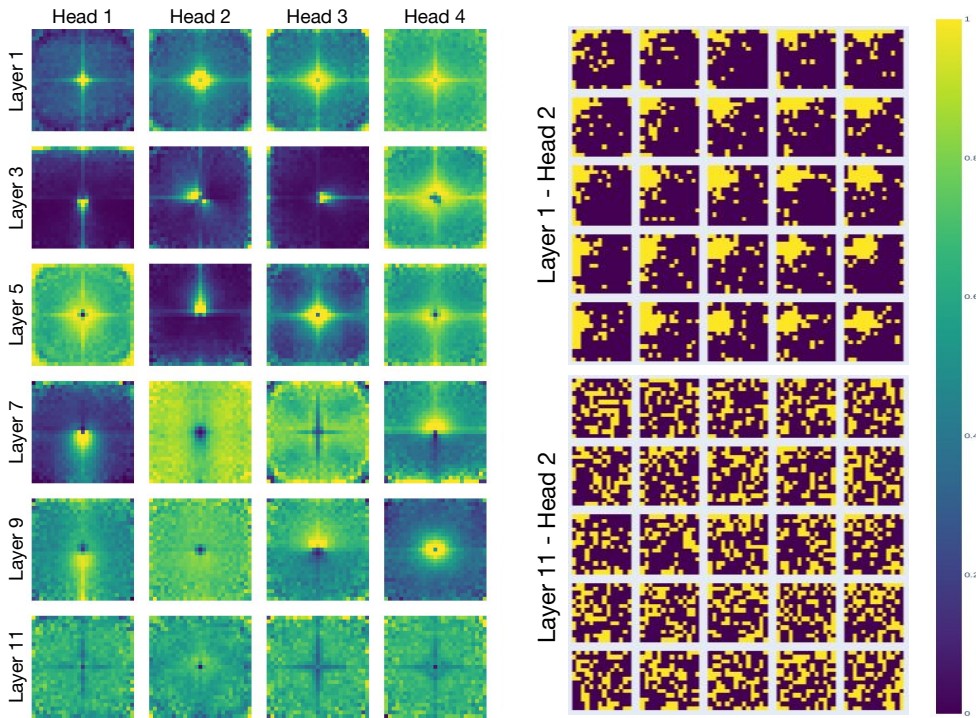

Figure 6: (Left) Query-centered mean of the magnitude of the priors $|\pi_{ij}|$ for each head in layers $\{1, 3, 5, 7, 9, 11\}$, learned from the ImageNet task. (Right) The binary pruning masks of the attention scores for attention score pruning via FiAK with pruning fraction 70%, showing the differences of the pruning masks between layer 1 and layer 11. Pruning masks in early layers capture local/short-range attentions while those in later layers capture non-local/long-range attentions.

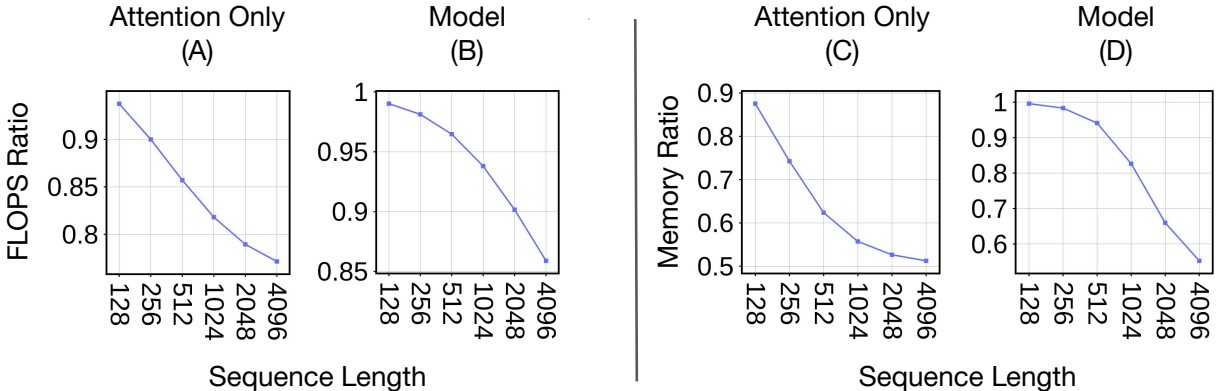

Figure 7: FLOPS and memory ratios at inference time on the WikiText-103 language modeling task for model pruned using attention score pruning via FiAK with the pruning fraction $k = 50\%$, compared to the baseline dense softmax transformer model. For a thorough analysis, we show a comparison at attention block only ((A) and (C)) and for the entire model ((B) and (D)). The advantage of the FiAK-based pruning grows with the sequence length.

# D An Analysis on Computational Complexity of the Pruned Models vs. the Dense Model

In this section, we compare the computational complexity of models pruned by our pruning methods with the dense softmax baseline. Following the same notation in Section 3 in the main text, we denote $H$, $D_x$, $N$, and $D$ as the number of attention heads at each layer, the input dimension, the input length, and the

model/feature dimension, respectively. To simplify the notation and computation, without loss of generality, we assume that $D_v = D$, i.e., the values have the same feature dimension as the queries and the keys. We also do not take the softmax operator into account. Since the linear projection of the H-head concatenated outputs is the same for the baseline and our pruned models, its computation is discarded for simplification.

**(i) Dense softmax attention:** The computational complexity for an H-head attention matrix is $N^2 H(4D - 1) + NHD(6D_x - 4)$.

*Explanation:* The output of a self-attention block at each head is computed via the following three steps (See Sec. 1.1).

- **Step 1** *Compute the matrices* $\mathbf{Q}$, $\mathbf{K}$ *and* $\mathbf{V}$ *via the linear transformations* $\boldsymbol{W}_Q$, $\boldsymbol{W}_K$, *and* $\boldsymbol{W}_V$. Since this step needs $3NDD_x$ multiplications and $3ND(D_x - 1)$ additions, the total computation is $3ND(2D_x - 1)$.

- **Step 2** *Calculate* $\mathbf{QK}^\top$. This needs $N^2 D$ multiplications and $N^2(D - 1)$ additions, thus $N^2(2D - 1)$ in total.

- **Step 3** *Compute the product* $\mathbf{AV}$. This requires $N^2 D$ multiplications and $N(N - 1)D$ additions.

Hence the total amount of computation for an H-head attention is $N^2 H(4D - 1) + NHD(6D_x - 4)$.

**(ii) Computation reduction of attention score pruning via FiAK**: Attention score pruned model via FiAK with the pruning fraction $k$ (See Algo. 1) has $kHN^2(2D - 1)$ less computations than the dense softmax attention.

*Explanation:* Attention score pruning via FiAK reduces the number of computation at step 2, i.e. calculating $\mathbf{QK}^\top$. Computations in other steps remain unchanged. Attention score pruned model with fraction of $k$ does not calculate the dot product of $k$ fraction of $(\boldsymbol{q}_i, \boldsymbol{k}_j)$ pairs, consequently saving $kHN^2(2D - 1)$ computations.

**(iii) Computation reduction of mixed pruning via FiAK**: Mixed pruned model via FiAK with the pruning fraction $k_1, k_2$ (See Algo. 2) saves
$2[(k_1 + k_2)D - k_1]HN^2 + (2D_x - 3)k_2 HDN$ computations.

*Explanation:* Similar to attention score pruned model via FiAK, at each attention head, mixed pruned model via FiAK with total pruning fraction $k_1$ saves $k_1 N^2(2D - 1)$ computation at step 2 above, i.e. calculating $\mathbf{QK}^\top$. Additionally, pruning $k_2$ fraction of $(\boldsymbol{k}_j, \boldsymbol{v}_j)$ pairs reduces computation at both step 1 and 3. At step 1, since matrix $K$ and $V$ accounts for $ND(D_x - 1)$ computations per head each, pruning $k_2$ fraction of the pairs saves a total of $2k_2 ND(D_x - 1)$ computations. Meanwhile cutting off $k_2$ fraction of $\boldsymbol{v}_j$ leads to $k_2[N^2 D + N(N - 1)D]$ computations for each head. As a result, mixed pruned model via FiAK saves a total of $2[(k_1 + k_2)D - k_1]HN^2 + (2D_x - 3)k_2 HDN$ computations for an H-head attention.

**(iv) Computation reduction of key pruning via GMM**: Key pruned model via GMM with the key pruning fraction $k$ (see Algo. 3) saves a total of $kHN^2(4D - 1) + (2D_x - 3)kHDN$ computations.

*Explanation:* As in mixed pruned model via FiAK, pruning $k$ fraction of $(\boldsymbol{k}_j, \boldsymbol{v}_j)$ via GMM saves $kND(D_x - 1)$ and $k[N^2 D + N(N - 1)D]$ computations at step 1 and 3, respectively. Moreover, for each head, pruning $k$ fraction of keys also saves $kHN^2(2D - 1)$ computations at step 2. In total, key pruned model via GMM needs $kHN^2(4D - 1) + (2D_x - 3)kHDN$ computations less than the dense softmax baseline.

Notice that the computation reduction is quadratic in the sequence length $N$. Therefore, when $N$ is large, i.e. long input sequences, the computational reduction achieved from using our FiAK-based pruning methods significantly increases.

