# OpenReview forum: "Pruning Transformers with a Finite Admixture of Keys"
_TMLR — Rejected by TMLR_

### Review · Reviewer_G9cm · 2022-09-15

**Summary Of Contributions:**

This paper proposes a novel probabilistic model for self-attention, namely the Finite Admixture of Keys (FiAK). It models the query distribution in self-attention as an admixture of Gaussian distributions centering around the attention keys. It is based on the finite admixture models (FAM), which is a generalization of finite mixture models (FMM) such as GMM. Specifically in FAM, the rows are drawn from different mixture distribution to provide more diverse attention patterns. Then, the experiments ablate the FiAK based attention-score pruning and mixed pruning, showing the method outperforms the GMM pruning baseline and other ViT pruning methods such as S2ViTE.

**Broader Impact Concerns:**

There are no concerns about the ethical implications of the work that would require adding a Broader Impact Statement.

**Requested Changes:**

Please provide more detailed analysis on the better representation capacity provided by the different prior distribution for queries in FiAK, and compare it with the modelling which models each key as a GMM. More implementation details of FiAK, i.e, how to jointly train and update the learnable priors, are recommended. Also, please add computation complexity analysis and comparison in the Tables, and comparison with more related works, such as SFW-pruning, minivit, tinyvit and others.

**Strengths And Weaknesses:**

Strengths:
(1)	The proposed FAM modelling has better representation capacity, due to its flexibility in choosing the mixture components.
(2)	Attention-score pruning and mixed pruning based on FiAK shows remarkable performance with large attention score pruning fractions (although less FLOPs reduction).
(3)	Detailed analysis and visualization of pruning masks illustrate the effect of FiAK.

Weaknesses:
(1)	Modelling self-attention as a probabilistic model was studied in (Nguyen et al. 2022). Besides connecting attention scores with GMM, it also models each key at a certain position as a mixture of Gaussians, which have a richer approximation. Compared with it, FiAK assigns different prior distribution for q, also aimed for better representation capacity. How much are the differences between the two design strategies and can FiAK capture more diverse attention patterns?
(2)	How do the learnable priors jointly train with the model? Please specify the details.
(3)	After pruning with FiAK, the model requires finetuning for a few epochs to recover the performance. However, an alternative is to train the model with sparse attention or use linformer from scratch. How does FiAK perform compared with these baselines?
(4)	Fig 2 provides the efficiency comparison for the experiments in Table 2. Can the figures also added to the table for a clearer comparison of the computational complexity and accuracy trade-off?
(5)	The number of compared methods is limited, which makes it difficult to judge the efficacy of the pruning strategy. Please add more related works and results comparison.
(6)	Computational complexity comparison is only provided in Table 3, please also add it to other tables.

---

### Review · Reviewer_oyMR · 2022-09-16

**Summary Of Contributions:**


Summary:
Thanks to the hard work of the dear authors, I have read this paper carefully. This article models the simple pruning problem into a relatively complex mixture of gaussian keys. The authors conducted experiments on several tasks. Overall, the authors' motivations and methods are OK, but the authors' experimental evidence does not adequately support their claims.


**Broader Impact Concerns:**


There is no concern about the ethical implications of the work.


**Requested Changes:**


Please refer to "Strengths And Weaknesses."


**Strengths And Weaknesses:**


(positive) The advantage of this paper is that it uses some new methods to solve the pruning problem. Since the advantages are apparent to all, I will not go into details.

(Negative) The authors' experimental evidence does not adequately support their claims. I am most interested in the results on ImageNet because this dataset is undoubtedly quite important. The authors have only experimented on DeiT-tiny, which I think is insufficient. I think the authors should do a lot of adequate experimental analysis on this dataset, as a lot of efficient architectural work does.

(Negative) The authors should publish results on ImageNet for more architectures, e.g., larger vision transformers. In the field of pruning, people sometimes like to see large models being pruned.

(Negative) Many papers achieve very high accuracy, even for small models at the DeiT-Tiny level. For example, DS-Net++ published in T-PMAI achieves a top-1 accuracy of 78.2% under the complexity of 1B-level MAdds. This accuracy is significantly higher than the authors' method (78.2% vs. 73.09%). Authors should compare or discuss SOTA methods (DS-Net++ is NOT a must. The authors can choose other SOTA methods for comparison.).

DS-Net++: Dynamic Weight Slicing for Efficient Inference in CNNs and Transformers

(Negative) Compared with other pruning works on ImageNet, the pruning rate of our method is too low.

(Negative) The authors are advised to report the physical runtime on ImageNet.

(Negative) In addition to the ImageNet dataset, the authors are also encouraged to report results on other datasets following the several suggestions above.


(Negative) While following the relevant literature, I noticed that this article was publicly reviewed. Among them, two impressive comments are that the effectiveness of the authors' method has not been fully experimentally verified. This includes some tables reflecting the authors' very insignificant gain on the LRA dataset and some figures reflecting the authors' very low pruning rates. Why did the authors remove these tables and figures? Please note that my rating is not based on these comments.

---

### Review · Reviewer_HkkA · 2022-09-17

**Summary Of Contributions:**

This paper presents a framework for pruning Transformers with a new FiAK method. Different from most existing methods that focus on removing redundant heads or tokens, the method is designed to prune redundant attention scores and keys. The method is evaluated on ImageNet classification, COCO detection and WikiText-103 language modeling tasks.

**Broader Impact Concerns:**

The method is designed to prune Transformers, which does not directly involve societal/ethical issues.

**Requested Changes:**

- The paper can be stronger if the results on more diverse model sizes and types are provided.

- It would be better to discuss or compare with the above-mentioned papers to highlight the contribution or advantages of this paper.

- The comparisons in Table 3 are not very clear since the best top-1 accuracy and FLOPs reduction are achieved by different methods. Is that possible to fix the FLOPs reduction to compare the top-1 accuracies? Presenting the results in a figure (e.g., scatter plot) may be helpful for readers to understand the results.

**Strengths And Weaknesses:**

Strengths:

- The proposed method is well-motivation and interesting, which can be a new method to reduce the computational cost of Transformer-based models.

- Experiments on multiple kinds of benchmarks (i.e., ImageNet classification, COCO detection, and WikiText-103 language modeling) show the method can outperform several baseline methods. It is also interesting to see the method is effective on both standard Transformers (DeiT) and hierarchical Transformers (Swin).

Weaknesses:

- The method is only evaluated on several relatively small models. It is still unclear whether the method is still effective on larger and stronger models (e.g., DeiT-Base).

- The paper didn't discuss or compare with some recent work such as [r1-r5] on pruning and reducing redundancy in Transformers.

[r1] AutoFormer: Searching Transformers for Visual Recognition, ICCV 20121.

[r2] Patch Slimming for Efficient Vision Transformers, CVPR 2022.

[r3] DynamicViT: Efficient Vision Transformers with Dynamic Token Sparsification, NeurIPS 2021.

[r4] IA-RED: Interpretability-Aware Redundancy Reduction for Vision Transformers, NeurIPS 2021.

[r5] A-ViT: Adaptive Tokens for Efficient Vision Transformer, CVPR 2022.

---

### Decision · Action_Editors · 2022-10-26

**Recommendation:** Reject

**Comment:**

The reviewers raised concerns about insufficient experiments and missing discussion of important prior work in the literature. The authors did not submit a rebuttal so all the concerns remain unresolved. To give a benefit of doubt, I read the paper and did not find strong reasons to overturn the reviewers' recommendations. Recommend rejection in its current form.

**Audience:**

Perhaps not strongly so. Network pruning is an important research topic that many people in TMLR's audience will find interesting, but with the lack of clear evidence in this paper, the claims and findings might be unconvincing to the TMLR audience.

**Claims And Evidence:**

The paper does not provide convincing evidence to support its claims. One of the critical concerns raised by the reviewers is insufficient experiments. In particular, they identified that the paper is missing an ImageNet evaluation and medium/large-scale Transformer results.